# Teaching Is a Story Whose First Page Matters—Teacher Counselling as Part of Teacher Growth

**Maarika Piispanen and Merja Meriläinen ***

Faculty of Education, Kokkola University Consortium Chydenius, 67100 Kokkola, Finland
* Correspondence: merja.k.merilainen@jyu.fi

**Abstract:** The starting point of the study is the work counselling process taking place in the induction phase of the students' teaching career. The work counselling process consists of activities that support and strengthen the students' teacher hood and their primary function as teachers by helping them to analyse their work and to attach to working life as newly graduated teachers. In this study, we use contextualisation and storytelling in the frameworks of positive pedagogy and systems theory to reflect on the students' growth process as teachers. We also acknowledge the importance of Vygotsky's zone of proximal development and, related to that, timely learning support (scaffolding) and a solution-focused approach as part of the supervision process utilising the methods of work counselling. In this study, autoethnography was used as a tool to help us reflect on our actions and work. Methods were chosen based on the thought that in the induction period, the students' experiences function as mirrors for us supervisors, helping us to reflect on our actions and the supervision process and to form ideas and meanings based on the documented discourse data.

**Keywords:** work counselling; scaffolding; teaching practice; peer supervision

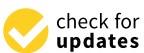

## 1. Introduction

In this study, the induction phase of studies is examined during the first months of the working life of a student transitioning from university to work. The transition from a pre-service teacher to a qualified teacher takes place during the fall when pre-service teachers work as substitute teachers in the Finnish primary school's first–sixth grade, as class teachers. We, the researchers, have taken part in several different teaching practice periods as supervisors. Traditionally, a supervision process consists of drawing up assignments for the students. The assignments are planned so that they would, in the light of theory, serve the students in the best possible way during their teaching practice and in their personal development as teachers. These assignments move the supervision process forward and function as starting points for the supervision. Most of the time reserved for supervisory discussions was used to go through the assignments. By doing so, we were able to make sure that the assignments were suitable and that the students were able to complete them at the desired level. For some students, reaching the objectives was easy, whereas some needed more assistance with the assignments. Part of the students have seen the teaching practice and the assignments as effortless, whereas some were stressed by the assignments because completing them while working was challenging.

As the students in the Kokkola University Consortium's teacher training program are adults with varying educational and work backgrounds, their individual experiences and their way of seeing themselves as teachers vary. This means that the transition from students to qualified professionals may take longer for some students, which should be considered even better than before in teacher training and especially in teaching practice.

In autumn 2020, we realised during a final teaching practice that as the students entered working life, they faced a lot of other central questions than those we had prepared them for through the assignments, even though we had chosen the assignments so that they

would strengthen the students as teachers. This was partly caused by the fact that during the supervisory discussions, we gave more and more room for the students' current thoughts about their work and themselves as teachers, but also used our own studies in the field of work counselling and therapy. We often wondered whether students entering working life would still need guidance in the assignments we had traditionally provided instruction for, or whether we should trust their growth process as part of the work counselling type of supervision process with which we supported the students in those teaching-related issues they found meaningful now. Finally, we decided to renew the teaching practice supervision process of a classroom teacher's adult education. The renewed supervision process was easily applicable in this framework.

Our work was steered by the assumption that a work counselling process would help us to offer more individualised and timely supervision and to make use of peer support in a more versatile way. A leading thought was also the idea that we as supervisors could stand by the students, compassionately supporting them in the challenging, demanding, and stressful induction phase.

This reasoning and several related discussions formed the research questions of this autoethnographic and discourse analytic study, which examines the transformation process of teaching practice supervision through our field notes. The focus of autoethnography is not the literal study of the self but the space between the self and practice. Autoethnography requires parity in the data gathered from the self and others as well as in how they are brought together to create meaning [1]. Because the personal is the domain of autoethnography, a study using this methodology provides evidence and analysis in research relevant to a context that extends beyond the reconstruction of a lived experience into the deeply personal and transformative [2]. The field notes work as narratives and tools of reflection [3] (p. 35). Autoethnographic text is characterised by self-awareness and awareness of impressive social environments [4]. Thus, in autoethnography, the researcher makes observations, which he reports in written form, for example. In these notes, i.e., field notes, we are looking for matters that are relevant and possibly relevant to this study. Clear research questions were a precondition for being able to discern from among all our experiences those that are related to our research topic and perspective and for being able to analyse the experiences. In addition to the research questions, we aimed to limit the experiences by writing down our impressions before and after the supervisory meetings and by recording the discussions and messages related to the meetings. The research questions are as follows: 1. What kind of discourses can be found in the field notes and discussions related to the work counselling process? 2. Can these discourses work as starting points for future supervision in the induction phase when the students enter working life?

## 2. Finding the Essentials with the Help of Autoethnography and Discourse Analysis

According to Lapadat [5] (p. 590) and Pink [6] (p. 122), focused notes produce special autoethnographic information: because of their reasoned and thorough nature, they provide a chance to analyse and examine everyday practices and societal and cultural structures and practices from a personal research perspective. In the study at hand, the topics of research are our personal development and experiences as supervisors in the work counselling process of the final teaching practice. The outcome is what we consider meaningful in the supervision process of student teachers in their induction phase.

When starting this research project, we had several years of experience in supervising students in their final teaching practice, developing the supervision process and overseeing the teaching practice. As described above, during the previous year's final teaching practice, we started a thought process and a dialogue about whether it would be possible to approach the supervision process from a new perspective. The extensive dialogue continued throughout the intervention, that is the final teaching practice, and we also kept a record of our thoughts and feelings. During the intervention, we felt we threw ourselves and our knowledge into the process: although we are experienced work counsellors, applying work counselling methods in the supervision process of a group of teachers who

were only just starting their careers and who were situated in different schools and work communities in different parts of the country really posed a challenge. On the other hand, we expected that collegiality would be the strength of such a group of students, uniting the students who were in the same situation despite their distance [7]. To give both of us supervisors a similar perspective to everyday work, both of us obtained the same number of students (*n* = 24) according to the age group of the pupils the students taught (grades one to four and grades five to six).

In autoethnography, it is essential that the researcher is part of the field instead of going to the field [8] (p. 79). Chang [9], who applied autoethnography himself as a teacher and educator, emphasises the role of the writing process at all stages of the method to allow analytical examination. This also leads to one of the major advantages of autoethnography in relation to other ethnographic methods: the data can always be reached by the researcher. Whenever and wherever he takes the observations and experiences made in the field again, he reworks his material, finding social structures from many sources of income. Austin and Hickey [10], who used autoethnography in the field of teacher training by working autoethnographically with pre-service teachers, found it presented valuable opportunities for its application in situations requiring a connection between self-understanding and broader socialisation processes. During the research process, we understood this autoethnographic process of ours as researchers, emerged to having pre-service teachers apply their own notions of self and understandings of identity and identity formation in a critical pedagogical way in their own professional practice post-graduation.

The suitability of autoethnography as the approach and data collection method of this research intervention was reinforced by the theoretical framework of solution-focused work counselling, which puts emphasis on the role of the counsellor as a supporter of the counsellee. In this way, the counsellor and the counsellee are equals, as the counsellor helps the counsellee to define his or her key objectives and goals. The counsellee, with his or her strengths and abilities, is the key person in defining the objectives and trying to achieve them. The role of the other students in the group is to support the counsellee's progress towards the objectives [11] (p. 3279), [12]. In a solution-focused work counselling process, the experience of an individual is always connected with a cultural experience, just like in the autoethnographic approach [12]. This could also be seen in the work counselling meetings where we shared our individual experiences and examined them through our shared cultural experience [13] (p. 739). Here, the relevance of culture emerged through factors shared by people in the same profession. These factors were related to professional values, identity, goals, skills, and abilities pursued through studies, range of practices, characteristic institutions, societal perspective, and the organisational system that is essentially steered by the Finnish National Core Curriculum for Basic Education [14]. The autoethnographic approach was also highlighted by the fact we were members of the work counselling group in the same 'culture' that we were examining [13] (p. 740).

This evoked ethical reflection since, as researchers and teachers, we were aware that we were exposing our work to the scrutiny of others at a stage where we were still contemplating the usefulness and quality of this new approach in the supervision process. This meant that instead of an outsider's description of the research topic, the research data included our feelings and experiences, even our uncertainty about the new approach [13] (pp. 13–16), [5,15]. We believe that the story-based approach of the study allows for empathy and identification, forming a connection between the researchers and the reader, which enables the formation of shared and common understanding based on an individual experience [16] (p. 262) [6] (pp. 64–65), [17] (p. 237). In this study, experience-based memories, diary entries, and recordings of our discussions and messages form the basis of the story-based approach. From a research perspective, these can be considered as quotations of the research data, which we examine with the help of discourse analysis. According to Kiviniemi [18] (pp. 79–82), even unofficial data such as an informal chat on a coffee break can be included in the research data. We also aimed at recording the key ideas of this type of discussion for the purpose of further research.

Discourse analysis proved to be an excellent method of analysis, as the object of study was our perceptions and related discussions. We wanted to examine the discourses that took place during the process and see whether they included shared and recurrent meanings. Pietikäinen and Mäntynen [19] (pp. 52–53) stated that the functions of discourse are like those of language: to describe the world and what takes place in it, to build one's identity in it, and to arrange resources that can give meaning to it. The data are used to analyse discourse: how something is described, and the meanings involved.

## 3. Results

In this autoethnographic study, the story is composed of the phases of the research process as well as the discourses taking place during the process. Therefore, it is central to view the research process from the point of view of the resulting story. The phases of the research process, the discourses arising during the process, and the feedback from the students form a whole that we examine in more detail in the result section of this study.

The research process can be divided into three phases, each of which produced extensive notes in our learning diaries and as text messages and recordings. In the next chapter, these processes are examined in more detail to achieve an overall picture of what our notes are based on.

### 3.1. Phase One—Key Factors in Constructing a Story

The first phase of the process consists of the construction of the final teaching practice study module. In this phase, our discussions focused on the learning processes constructing the teaching practice and the learning assignments involved. What is central, necessary and significant when guiding students to attach to their work and to develop themselves as teachers? What kind of processes and assignments involved in the processes will support their everyday coping and strengthen their professional growth and self-efficacy? This can be described with the help of a book metaphor: if teaching was a story, what should happen on the first pages, so that one could follow the plot?

Phase one was backgrounded by our experiences of supervision processes from the previous years: in these processes, we concentrated on guiding and supporting the students in completing their traditional learning assignments in the right way. Since the timeliness of guidance and support as well as students' individual needs which stem from their diverse life situations had become more meaningful to us as supervisors, we decided to develop the teaching practice supervision process to the direction of work counselling. This way, the students themselves could decide which questions related to their professional growth we would discuss. They could also be supported by the other students in their stage of professional growth as teachers. Here, we wanted to highlight the dynamic nature of resilience: when individuals support one another with empathy and as equals, both the supporting and the supported person's resilience is strengthened [20] (p. 1652), [21] (p. 185).

We decided to carry out a semi-structured work counselling process where a student teacher's professional growth was supported by current articles and webinars which the students could familiarise themselves with independently during the different phases of the process. The thoughts provoked by the articles and webinars would be used in the work counselling process. It is characteristic of a work counselling process that there should be time and room for the counsellee's individual questions and needs for support.

Learning assignments that were meant to support the teaching practice process now worked as tools for the process, leading the students to reflect on themselves as teachers and stirring discussion about each student's current learning needs in relation to teacher competencies. Pantic and Wubbels [22] divided these competencies into four categories: values and raising children; understanding of the school system and participation in developing it; subject knowledge, pedagogy, and curriculum; and self-evaluation and professional development. Soini, Pietarinen, Toom and Pyhältö [23] divided the same things into the following categories: (1) knowledge of learning, (2) interaction skills, (3) well-being

skills, and (4) knowledge of school development. According to them, these dimensions of knowledge are central in teachers' main professional contexts, namely classroom interaction and the professional community. However, Heikkinen, Aho and Korhonen [24] specified that teachers' support needs are different in the different phases of their careers. This made us ponder which questions we should address during the structured part of the supervision process. According to Soini et al. [23], at the beginning of a teaching career, teachers need support in further developing the basic knowledge gained from teacher training and in many practical issues related to teachers' work. Bearing this in mind, we chose the following themes for the assignments, articles, webinars, and other material used in the supervision process:

1. Drawing up plans (term plans and course plans to support teaching);
2. Operational culture, theory-in-use and classroom practices;
3. Cognitive control, respecting one another, goal-oriented activity;
4. Strengths and resources and how to utilise them;
5. Resilience and how to strengthen it.

These themes worked as the students' professional development tools during the teaching practice and they were discussed during the structured part of the work counselling process as students raised questions and ideas related to them [11] (p. 3283), [25] (p. 57). As for the open phase of the work supervision process, we wanted to allow room for students' topical issues.

Students' attachment to working life was also supported by the work community: each student had a personal tutor teacher whose task was to support the student in attaching to working life and school culture. The cooperation between the tutor teachers and the students was also supported during the supervision process.

In the online learning environment of the teaching practice, the initial words are as follows: "Take with you educational science–the zone of proximal development, and the power of timely support, i.e., scaffolding..." We also put our trust in these words in the work counselling process, where the students themselves were given the chance to define the central objectives for the supervision on the basis of their support needs [26] (pp. 43–44). As our approach to work counselling is both resource-focused and solution-focused, we decided that the supervisory discussion would be held in groups. Collegial group discussions utilising peer support would enhance the use of listening, enquiry methods, clarification, and projection, as well as elements affecting one's thinking. The following extract, from a recording of a discussion, illustrates our contemplation:

> **Supervisor A:** *From a work counselling perspective, would it not be rather bold to ask the students which questions they would like to address today? They might find topics for discussion from the webinar they have listened to, but if they had something else on their minds, we could give room for that as well. This would be a solution-focused approach that would help us tackle matters presented by the students in a timely manner.*

> **Supervisor B:** *That is a bold suggestion, but it would also involve timeliness, the zone of proximal development and always scaffolding. It would enable the development of teacherhood in one's own zone of proximal development and, at its best, make it possible to receive collegial support from others who are probably in a similar situation or who have already processed the situation.*

In the first phase of the planning process, we felt both enthusiastic and uncertain, which can be seen in the following discussion between us:

> **Supervisor B:** *The uppermost feeling is enthusiasm towards the new model and the lately developed awareness of the importance of timely support. The new approach is also supported by my own experiences of the solution-focused approach and of the empowering nature of positive psychology in work counselling.*

> **Supervisor A:** *There is also excitement in the air, caused by the following question: How will the students, who are used to completing assignments in accordance with a*

*certain protocol, react to supervision that is based on each student's current questions and objectives? Will the students be able to make good use of the supervision and really highlight their current supervision needs as regards their teacherhood? Will I be able to address questions that I have probably not been able to prepare for beforehand?*

*3.2. Phase Two—The Plot of the Story Is Composed by the Students*

The second phase of the research project is related to the implementation of supervision. We see this phase as an intervention phase involving work counselling meetings. The work counselling process consisted of weekly group meetings (7 weeks), for which the students (*n* = 24) were divided into two groups as described earlier. As the group consisting of student teachers working in the fifth and sixth grades was slightly larger, it was further divided into two groups for some of the meetings. This way, each student was given more possibilities to join the discussion. Thus, we met each student seven times during the work supervision process (six group meetings and one one-to-one meeting). When planning the work counselling process, we utilised models by different researchers [27] (p. 79), [28] (p. 194). By combining these models, we ended up with the following structure for the meetings:

- Reinforcing pausing and presence;
- Addressing basic questions;
- Discussing the objectives;
- Summary and conclusion.

It is important that counsellees can stop and settle down at the counselling meetings so that they can make use of the work counselling in the best possible way and be present for themselves [27] (p. 79). Pausing, peaceful music, a poem or an aphorism were excellent ways to start the work counselling because usually, the students come to the meeting straight from their classrooms. Basic questions, on the other hand, helped to keep the meetings goal-oriented and they also helped the group members to identify and support the growth of one another toward the objectives [28] (p. 194). Before the work counselling process began, the students discussed the objectives in the online learning environment of the teaching practice, where the students wrote their thoughts and questions about work situations that they had faced during the previous week and which they wanted to discuss during the meetings. As there is limited time for the meetings and effective supervision requires focus, it is often necessary to concentrate on certain issues. This means choosing a topic for the meeting and sticking to it. It is the work counsellor's task to help the counsellee to choose a topic and to see that the discussion and other work counselling methods concentrate on dealing with the topic carefully enough.

When concluding the meeting, all persons present need to have a shared view of the topic and focus of the meeting. The focus determines from which perspective issues included in the conclusion are viewed and selected. The conclusion also works as a way to show that the supervisor is actively listening to and understanding the counsellees, which leaves them with a pleasant feeling of having been heard.

In between each meeting, the supervisors contemplate the topic of the next meeting—especially the basic questions, which are central in proceeding towards the questions and discussion topics received from the students. We also focused on pondering how we could highlight the perspective of collegial and shared expertise in the solution-focused approach, where the supervisors do not give ready answers but instead help the students to find the solutions to different situations themselves and with the help of peer support.

> **Supervisor B:** *I participated in the discussion a few times, but I do not think I gave any answers–we did not talk about plans at all, which is why it was easy to keep summaries and conclusions to a minimum.*

This research phase was very multidimensional: we had discussions with the students, both also with each other as colleagues and with the theory. The central themes arising from the theory were methodological approaches such as using the solution-focused ap-

proach [12,29,30] to positively reinforce collegial support. A strong factor in supervision is self-efficacy. Ryan and Deci's [31] self-determination theory supported this and the importance of work counselling discussions when self-efficacy is strengthened in the framework of collegial support. Self-efficacy involves that the more a person believes in their ability to succeed, the better their motivation to work towards their goals. As work counsellors, it was our task to help the students to observe their earlier success, to be interested in their strengths, and to look for positive exceptions in discussions. Improving one's self-efficacy is directly related to the strengthening of one's agency, which can be detected in the objectives of the final teaching practice.

According to Ryan and Deci [31], intrinsic motivation is particularly based on satisfying the three main psychological needs, namely competence, autonomy, and relatedness, which also are essential factors in the attachment of a person to their profession at the beginning of their career. According to the self-determination theory, to feel intrinsically motivated, individuals must feel that they are self-efficient and capable. In work communities, the feeling of self-efficacy can be strengthened when people feel that they can manage challenging situations either because they are capable or because they can obtain collegial support. According to Ryan and Deci, intrinsic motivation also requires the experience of autonomy.

When aiming to find solutions through collegial cooperation, we also saw Seligman and Csikszentmihalyi's [32] idea of well-being as central. According to them, well-being is not only the absence of malaise but involves separate independent phenomena that need to be examined as much as problems. Therefore, one of the key starting points in our work counselling discussions was trying to find solutions to challenges together, identify success and grow professionally through shared expertise and prevention. Can we, together, learn to share our everyday life and find factors that reinforce our profession?

> *Supervisor B: Old-fashioned practices and the lack of resources and materials was realism in the daily life of many students, so I instructed them to profoundly contemplate on their philosophy of education: What is possible even though you are lacking something? That started a good discussion about how easy it is to find reasons for not doing something and in a way sell one's teacher hood. Instead, one could think about what one can do despite shortage of materials and how one could improve the resources and materials of the school step by step.*

> *Supervisor B: The stories were touching and beautiful and conveyed the feeling that one's work is meaningful: interaction, encounters, providing a sound basis for life, and producing shared memories were the common thread of being a teacher, and the examples of these were from daily life and meaningful encounters.*

> *Supervisor A: Today we discussed broadly about what kind of a teacher one would like to be and what is meaningful in one's teacherhood. Quite a many students considered that meaningful encounters are important. I noticed that it was not easy to verbalize this: How do meaningful encounters show in my daily life? Many said great things about encounters but were not able to give concrete examples. We discussed this for a long time. The students were perplexed, too, about how difficult it was to verbalize one's actions even though that is one of the core tasks of a teacher.*

> *Supervisor B: The assignment of the day–expressing one's way of being a teacher in words–made everyone focus on the same issue in an atmosphere that was clearly meaningful. The students verbalized their thoughts very profoundly, and the respectful way they spoke about pupils and encounters with parents made me feel that it was a group of experts, not a group of students working on an assignment.*

Our field notes conveyed the various dimensions of meaningful encounters: those between colleagues, between the supervisor and the group, encounters with pupils and with one's work community, as well as facing yourself in the situation as such and from a teacher's point of view. The students clearly needed guidance for pausing in these encounters because the different dimensions of encounters are blended in daily life, and it

is challenging to examine one's work and actions from different perspectives unless one can distinguish between the perspectives.

In terms of a story, this was a plot-creation phase: as the students' needs were given room from a solution-focused perspective, we could not always anticipate the turns of events.

The time used for handling the topic of the day depended on the discussions we had on the topic. As supervisors, we maintained and enhanced the discussion whenever necessary. Most of the time, the students were very quickly absorbed in the discussion and discussed the topics by offering observations and questions regarding their daily life, which the other students could easily relate to.

> **Supervisor A:** *I am happy that the students are so open when discussing "the burning issues of the day": They are open about their uncertainties and situations that they have managed well. It is great that the students support one another and help each other to analyse their thoughts.*

Even after just a few work counselling meetings, we felt that the students had anticipated our meetings and had even more courage to participate in the discussions also by revealing their own uncertainties to receive support from others. This brought us a lot of joy because it showed us that we were reaching the goal we had set for the supervision process. We observed that the students had also created their own networks to support one another:

> **Supervisor B:** *Professional attitude and appreciation of one's work were palpable when the students discussed current school-related questions, and we paused for a moment to consider authenticity as initiator of motivation. This had a firm basis on correspondence between the pupils of three students and the pupils' enthusiasm about it.*

> **Supervisor A:** *I sense that the students are looking forward to our meetings and are well-prepared for them. Some sit on the sofa with a cup of coffee, some have a pen and a notepad, some have a list of questions they wish to address together with their colleagues. The students have also started to talk about 'work counselling' instead of 'practice supervision'.*

> **Supervisor B:** *This time the discussion was lively and spontaneous, and I heard the phrase "I will use that in my class" several times. The students also openly gave one another feedback and praise when they felt good about what somebody said or felt it was thought-provoking.*

Instead of the role of a student, the students seemed to take on the role of a teacher: the discussion is professional and collegial, with an atmosphere of shared expertise. As supervisors, we were able to step aside from the supervisor role, which is central to the solution-focused approach [11,21].

### 3.3. Phase Three—Various Interpretations of the Story Form a Shared Experience

The third research phase consisted of individual work counselling meetings that were based on the students' learning diaries, of which they had compiled syntheses for us. We asked them to put together those factors in the teaching practice that were centrally related to their growth and development as teachers and that they would like to discuss with us in person. In addition to the syntheses, the students also gave us anonymous feedback on each work supervision meeting and on the teaching practice as a whole.

Reading their syntheses and feedback was in a way very exciting because, so far, our field notes had consisted of our own interpretations based on our experiences and feelings: now we would see each student's view of their development during the practice and the anonymous feedback would provide us with direct feedback on the practicality of this type of teaching practice. The students' documentation could either confirm our interpretations or prove our hypothesis on the practicality of a solution-focused work counselling process

wrong. In that case, the discourses in our discussions would be misinterpretations of the meaningful factors in a supervision process.

Even though we are not able to use the students' syntheses as such as research data, we use the syntheses as a tool to examine and reflect on the discourses in our communication and notes. This can be considered as a central factor from the perspective of reliability: do the students' syntheses and feedback confirm the central discourses we found in the research data? Naturally, the concept of orthodox reliability cannot be applied in this ethnographic research, where we use the research data to create our personal view of how the students experience the supervision process. We aim to express our experiences in words, which is always the truth told, not the event itself [13] (pp. 745–751).

## 4. Discourses and Their Interpretation

We used the concept of discourse to analyse the supervision process. Here, the concept of 'discourse' is used to refer to a meaningful relationship or dimension that occurs in our texts repeatedly and constructs our conception of the supervision process, its relations, and meaningful factors. The context in which we use language gives the words and sentences their own meaning, and meaning systems produced with the help of language are best understood in the contexts they take place in. The direct quotes in the study are taken from our discussions and notes and are our personal interpretations of reality. These quotes have a situational function in the context, time and place they occur [33] (p. 18).

The following six discourses were central to the study: timeliness, meaningfulness, encounters, collegiality, solution-focused level, and shared expertise. These discourses form parallel meaning systems, as it is difficult to be aware of all the meanings we produce. In the table below (Table 1), we describe the discourses taking place during the research process and the meanings linked with them in this context. To confirm the reliability of the interpretations, we also included comments from student feedback in the table.

**Table 1.** Six main research discourses.

| Discourse | Meanings | Examples from Researchers' Notes | Examples from Student Feedback |
|---|---|---|---|
| timeliness | timely support (scaffolding) | "That would mean utilizing timeliness, zone of proximal development and always scaffolding." | "Today I especially needed support for my well-being and coping." |
| meaningfulness | experiences of meaningfulness | "The stories were touching and beautiful and conveyed the feeling that one's work is meaningful." | "These moments are very important for the beginning of my career." |
| encounters | willingness to face others genuinely | "...the respectful way they spoke about pupils and meeting with parents made me feel that it was a group of experts, not a group of students working on an assignment." | "I think our discussion about the disadvantages of the profession has been very honest, respectful and open during the work counselling meetings." |
| solution-focused level | giving guidance without giving answers | "...instead, one could think about what one can do despite shortage..." | "During the practice period, reinforcement was provided for issues that were relevant to me." |
| collegiality/shared expertise | -support expected from colleagues -sharing the daily life -sharing ideas | "... they openly talk about their uncertainties and situations that they have managed well." | "These moments are very important for the beginning of my career. I am still so uncertain of the things I do. Encouragement and peer support are always useful." |

The existence of the central meanings found in the discourse analysis becomes apparent when they are examined through the phases of the research. The discourses can even be considered presumable, but on the other hand, their multidimensionality would not have become visible without an exploratory study and analysis. From the perspective of

discourses, it is necessary to examine the phases of the research as they, in part, explain our story and our choices in constructing the work supervision process.

The multidimensionality of meanings reinforces the challenges and possibilities related to the ambiguity of language, which became apparent in this work supervision process in many ways. When the students talked about 'encounters' or 'meaningfulness of work', for example, they had difficulties at first in describing what these concepts mean in everyday actions, or what they are composed of. These concepts could be described as collective and comprehensible to everyone but the individual experiences related to the concepts can give the concepts different meanings in different peoples' minds [34] (pp. 81–82). As supervisors and teachers, we learned how important it is that we explain the concepts used and ask the students to describe the concepts they use in more detail and from the point of view of their work. If the concepts are only seen as abstract concepts and their meaning in practice is not understood, the work community may lack a shared understanding of the actual meaning of a matter. The same is true in the context of teachers and pupils: explaining the concepts used is a key to achieving the wanted activity and results. The systemic whole of language is linked to the meanings of the concepts used [35] (p. 120).

Each person constructs their selfhood in the social context they live in. Together, people construct the human world and its socio-cultural and psychological constructions [34] (pp. 61–63). Language creates reality and relationships between people, thus being an important tool in cooperation. Even from this point of view, supervision that utilises the methods of work supervision and that aims to be solution-focused is justified especially in a phase where collegial cooperation between newly graduated and newly employed teachers, who are all in the same situation, can share their experiences and expertise. We would probably not have reached such a high level of collegiality, shared expertise and timeliness without approaching the work counselling process from the point of view of a solution-focused approach.

Examining discourses from the perspectives of social construction and interaction is an old field of study but present in our everyday life. These group discussions utilising methods of work counselling reminded us of the importance of sharing and interaction. During work counselling discussions, the discourses expanded into expanses of talk and formed a professional discussion including expressions that connected the participants in a professional sense and formed a vaster discussion whole [36] (pp. 54–55). The communal construction of the existing and shared knowledge and an experience that unifies the profession both strengthens, shapes and changes the institution and the jointly built professional identity, in this case, teacher hood, from the perspectives of shared expertise and collegiality [34]. When analysing the discourses of collegiality and shared expertise, we observed that they were so closely linked to one another that we decided to join these concepts.

The significance of timeliness often comes up in our discussions, in the framework of the solution-focused level and from the point of view of scaffolding. The starting point of the solution-focused approach is that the supervisor helps the people supervised to find the solutions themselves by asking 'timely' questions which advance the situation [12]. As supervisors, we had a central role in the supervisory discussions.

On the other hand, the webinars, articles and materials also guided the students to examine current teaching-related themes in a structured way, thus promoting their professional growth. We were able to provide timely support in each student's zone of proximal development during the entire supervision process by giving room for the students' questions and thoughts. We see that collegial guidance was of great importance here: the students were supported in their questions both by their fellow students and by us supervisors, as we tried to leave room for peer supervision. This way, we also aimed at strengthening the professional identity of these graduating teachers and their ability to have collegial discussions in their workplaces. This approach supports studies showing that interactive collegial support strengthens the resilience of both; the supporter and the person receiving support [20] (p. 1652), [21] (p. 185). As we examined the objectives of the

teaching practice and discussed them with the students, we became convinced that our way of supervision works in improving professional growth and resilience.

## 5. Conclusions

> ***Student feedback:*** *I feel that during this practice period I have been safely seen off at working life. It has been really wonderful to assemble with my fellow students once more to discuss the challenges of working life. It was also comforting to notice that I am not alone with my anxiety, and that many things are actually fine. The webinars and other assignments were also good and supported my work. Thank you for seeing me off.*

Even though our study focused on the analysis of our own field notes, which is typical of the autoethnographic approach, we felt that the students' syntheses supported our decisions. We felt that student feedback supported the research result that utilising the methods of work counselling in the supervision process, especially in the induction phase, enhances and supports attachment to the job, although it requires strong professional skills in the methodical approach, as well as sensitivity and plunging into the authentic situation, which often involved questions and needs about unexpected themes. It is central to identify the role of the work counsellor from a supervisor's perspective, where development and developing are based on learning by analysing one's own and fellow students' experiences in interaction with the supervisor and each other [37] (p. 22).

Throughout the research process, we reflected on one question: is our way of doing things acceptable? What is interesting in this question is that although we both are experienced work counsellors and therapists, in most of our discussions we aimed at reinforcing our way of operation. This type of action could also be called uncertainty. Afterward, we observed that our uncertainty did not arise from a lack of competence or from nervousness caused by different situations but from a new way of doing things. It was something that we had not carried out before when supervising students in their final teaching practice. It had to do with change and departing from tradition, although we did not resist change. We only needed to ask the following questions: Do we have the courage to break tradition and do something that has been preserved for a long time? Do we have the courage to trust that people have individual needs that must be observed in a timely manner so that support can be directed at the current needs so that students' professional growth would continue and not come to a standstill? [38] (p. 534).

Throughout the research process, we obtained support for our thoughts from one another, the students participating in the supervision process, and studies related to the topic. Afterward, when examining the discourses and the results derived from them as well as the student feedback, we can agree with Ryan and Deci [31] according to whom the social relationships and environment provided by work counselling are factors that have a significant impact on individuals' emotional experiences throughout their lives. Individuals have a fundamental need to feel part of and supported by their community. This need becomes even more obvious when entering working life. According to Ryan and Deci [31], a lack of social support and acceptance easily leads to the decline of feelings of autonomy and competence, which can indicate burnout or feelings of inability, inadequacy and incompetence, and/or the lack of attachment to work. Thus, in their theory of self-determination, Ryan and Deci [31] highlighted the importance of intrinsic motivation as a central factor in improving psychological well-being. We need each other to be mirrors that show us that we are not the only ones to feel uncertain in a situation and also that we are not alone in that situation.

In traditional autoethnography, the meanings in the research data are different for the research subjects, who are in the middle of the events, and the researcher who observes the events from outside. In autoethnography, the roles of the research subject and the researcher overlap. Thus, meanings arise from two contexts: the original context of the research data and the autoethnographic writing and interpretation [39] (pp. 127–128). As researchers, we see that as there were two of us, this phase was slightly easier: although we both made our own notes, we discussed them as researchers. We feel that the students'

syntheses and feedback supported our interpretation in a significant way: our view of the importance of utilising work counselling methods in teaching practice supervision was shared by the students.

All in all, the results reinforced our assumption that work counselling can be used in the supervision of final teaching practice, especially when the teaching practice is instructed and takes place at the workplace in a phase when the students are entering working life and doing their advanced studies. The research also gave us confirmation and courage to further develop processes of teaching practice supervision in teacher training by utilising methods of work counselling, even if work counselling methods are rather rarely used in universities' practice supervision. This can be explained by the fact that work counselling is a demanding method that often requires that the supervisor is trained as a work counsellor or has equivalent training in the methods of work counselling. On the other hand, the results and the student feedback made us consider whether universities could provide work counsellor training that could be applied in practice supervision and would enhance professional development.

Autoethnographic research is sometimes considered inadequate: it is said it is not detailed, theoretical or analytical enough. On the other hand, it may be considered too aesthetic, emotional, or therapeutic [12,40]. When contemplating our study, we can state that research can be all this at the same time and combine personal and social dimensions in a thought-provoking way: this is what the supervision process utilising work counselling methods achieved [40] (p. 11). The aim of autoethnographic research is to produce socially equitable, analytical, and accessible texts that can change us and the world around us for the better [12] (p. 764). The student feedback made us feel that the supervision process both strengthened the teacher hood of these teachers who would soon be entering working life and enhanced our own professional growth as teacher trainers and work counsellors at the same time.

**Author Contributions:** Conceptualization, M.P. and M.M.; methodology, M.M. & M.P.; software, M.P. & M.M.; validation, M.P. & M.M., formal analysis, M.P. & M.M.; investigation, M.P. & M.M.; resources, M.P. & M.M.; data curation, M.P. & M.M.; writing—original draft preparation, M.P. & M.M.; writing—review and editing, M.P. & M.M.; visualization, M.P. & M.M.; supervision, M.P. & M.M.; project administration, M.P. & M.M.; funding acquisition, M.P. & M.M. All authors have read and agreed to the published version of the manuscript.

**Funding:** This research was funded by Jyväskylä University. 819/12.00.00.03/2021.

**Informed Consent Statement:** Informed consent was obtained from all subjects involved in the study based on the ethical principles of research with human participants and ethical review in the human sciences in Finland Finnish National Board on Research Integrity TENK guidelines 2019.

**Conflicts of Interest:** The authors declare no conflict of interest.

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
