# Peer review of "Teaching Is a Story Whose First Page Matters—Teacher Counselling as Part of Teacher Growth"

_education, doi:10.3390/educsci12120862_

Round 1
Reviewer 1 Report
This is an interesting study which I’ll try to briefly describe here so that the authors can judge whether I have understood it correctly. As I understand it the study focusses on how early career teachers are supported to develop their professional identities, with the authors taking an autoethnographic approach to explore an innovative approach they trial in their own practice as teacher educators or the equivalent. This is an important and potentially powerful story: not just the supported offered to the beginning/student teachers, but also the bravery of changing practice and how it made the authors feel.
The context of the study is not fully explained and so it feels like the reader is left to figure some of this out for themselves.
There are two potentially valuable contributions from this study, both of which need more information to fully explain:
1. A critical review of autoethnography as an approach to researching our own practice in education;
2. An approach to (better) supporting teachers in the challenging early stages of their careers.
To improve this article, I offer the following suggestions for revisions:
- Provide the context for the study more fully: where is it taking place, what is the standard/expected practice for supporting early career teachers, what subjects/pupil age groups do the teachers work with, are the teachers qualified or still classed as ‘student teachers’ etc?
- Give an overview of the existing literature about support for early career teachers, particularly that offered by university-based teacher educators, so that this approach sits alongside what is already known about early career teachers’ learning needs.
- Say more about the use of autoethnography in education research: such as where else has it been used, what are its benefits and drawbacks?
- Say more about the ethical issues as they relate to this study. Perhaps this study did not receive or need ethical approval, but this can be explored in relation to the literature about autoethnographic approaches.
- Further, in relation to ethics, the authors do use some student teachers’ quotes: did the teachers give permission for these to be used?
There are some other points which could be addressed:
- The extracts of dialogue do not read like speech. I assume they’ve been translated into English and so have lost the ‘flow’ of natural speech. Can the authors acknowledge this, since when I came to reading them I wondered whether they had come from written text rather than speech?
- I felt that the first sections of the findings actually describe the context of the study and so should be moved to a different ‘context’ section.
- There is a quote on lines 208-210 which isn’t cited, and I didn’t fully grasp its purpose or meaning.
- The authors might want to make reference to literature around communities of practice or professional learning communities, both commonly used in relation to teacher professional learning.
- The use of ‘students’ throughout is confusing: did the authors work with qualified teachers, or student teachers?
- Line 29 “forward” is not a verb.
- Some references have page numbers where they are not needed.
- Reference [9] (line 114) possibly doesn’t refer to the right text?
- Ryan and Deci line 292 needs a reference [27]
Author Response
Thank you for the review and the valuable comments for our text. Please see the attachment and find the latest version here.
Yours sincerely / Authors

Reviewer 2 Report
Dear authors, congratulations on your work. It was a good read!
Please consider typesetting revisions: use bullets or numbers with consistency, mark the double space in the text, and replace them with single space.
As for the content, consider adding headings such as Research limitations and Future research.
Author Response
Thank you for the review and the respect for our work. Please see the attachment and find the latest version here.
Yours sincerely /Merja and Maarika
